# Retinal Neuron Is More Sensitive to Blue Light-Induced Damage than Glia Cell Due to DNA Double-Strand Breaks

**DOI:** 10.3390/cells8010068

**Published:** 2019-01-18

**Authors:** Pei Chen, Zhipeng Lai, Yihui Wu, Lijun Xu, Xiaoxiao Cai, Jin Qiu, Panyang Yang, Meng Yang, Pan Zhou, Jiejie Zhuang, Jian Ge, Keming Yu, Jing Zhuang

**Affiliations:** State Key Laboratory of Ophthalmology, Zhongshan Ophthalmic Center, Sun Yat-sen University, Guangzhou 510060, China; peichen912@163.com (P.C.); LaiZhp3@mail2.sysu.edu.cn (Z.L.); wuyh45@mail2.sysu.edu.cn (Y.W.); xlj727@163.com (L.X.); caixx1987@126.com (X.C.); qiujin19940916@163.com (J.Q.); yangpanyang9@sina.com (P.Y.); 18819253882@163.com (M.Y.); zhoup23@mail2.sysu.edu.cn (P.Z.); zhuangjiejie@aliyun.com (J.Z.); gejian@mail.sysu.edu.cn (J.G.)

**Keywords:** blue light, retinal neuron, glia cell, DNA double strand breaks

## Abstract

Blue light is a major component of visible light and digital displays. Over-exposure to blue light could cause retinal damage. However, the mechanism of its damage is not well defined. Here, we demonstrate that blue light (900 lux) impairs cell viability and induces cell apoptosis in retinal neurocytes in vitro. A DNA electrophoresis assay shows severe DNA damage in retinal neurocytes at 2 h after blue light treatment. γ-H2AX foci, a specific marker of DNA double-strand breaks (DSBs), is mainly located in the Map2-posotive neuron other than the glia cell. After assaying the expression level of proteins related to DNA repair, Mre11, Ligase IV and Ku80, we find that Ku80 is up-regulated in retinal neurocytes after blue light treatment. Interestingly, Ku80 is mainly expressed in glia fibrillary acidic protein (GFAP)-positive glia cells. Moreover, following blue light exposure in vivo, DNA DSBs are shown in the ganglion cell layer and only observed in Map2-positive cells. Furthermore, long-term blue light exposure significantly thinned the retina in vivo. Our findings demonstrate that blue light induces DNA DSBs in retinal neurons, and the damage is more pronounced compared to glia cells. Thus, this study provides new insights into the mechanisms of the effect of blue light on the retina.

## 1. Introduction

Blue light (450–495 nm) is a major component of visible light and ubiquitous in the modern age because it is widely used for many technologies, including digital screens (TVs, computers, laptops and smart phones), electronic devices, and fluorescent and LED lighting [1]. The human retina is protected by the cornea and lens. However, blue light could penetrate the cornea and lens and reach the retina. Although an appropriate amount of blue light has a suppressive effect against myopia [2], over exposure to blue light could be harmful to retinal cells, causing retinal pigment epithelial (RPE) impairment, photoreceptor degeneration, and retinal ganglion cell (RGC) apoptosis [3,4,5]. Currently 60% of people spend more than 6 h a day in front of a digital device. Therefore, both epidemiological and animal studies indicate that direct and long-term blue light exposure is a co-factor in many retinal degeneration diseases. For example, constant exposure to blue light has been linked to the development of age-related macular degeneration [6]. However, the mechanism of its damage is not well defined.

Previous studies demonstrated that several mechanisms are implicated in the pathogenesis of blue light-induced cellular damage, including mitochondrial dysfunction, photo-oxidative stress accumulation and apoptosis activation [7,8,9,10]. The severity of damage varies depending on the light source, light intensity, or exposure duration [11,12,13,14]. The studies of Huang et al. showed that blue light exposure affects mitochondrial function, induces the generation of reactive oxygen species (ROS), and subsequently, induces cell apoptosis, using a RGC-5 cell line in vitro [15,16]. However, most studies were conducted in immortalized photoreceptor precursor cell line, which is different from primary retinal neurocytes. Moreover, the vertebrate retina contains several major cell types, which could be divided into proliferative cells and terminally differentiated cells. Blue-light induced damage might be different in different cells. Therefore, more investigation is required.

It is commonly recognized that DNA instability and DNA repair deficiency have been implicated in the initiation and progression of retinal neurocyte degeneration [17,18]. The accumulation of DNA breaks induced by pathological factors typically occurs in neurocytes [19,20]. Sasaki’s study demonstrated blue light induced DNA breaks in retinal cells both in vitro and in vivo [21]. However, DNA damage induced by blue light and its repair mechanism in proliferative cells and terminally differentiated cells remains unclear. Therefore, in this study, blue light exposure models both in vitro and in vivo were established to explore photochemical DNA lesions in proliferative and non-proliferative retinal cells.

## 2. Methods

### 2.1. Ethics Statement

This study was approved and monitored by the Institutional Animal Care and Use Committee of Zhongshan Ophthalmic Center (Permit Number: SYXK (YUE) 2010-0058), and strictly complied with the ARVO Statement for the Use of Animals in Ophthalmic and Vision Research. Sprague Dawley (SD) rats were obtained from the Ophthalmic Animal Laboratory, Zhongshan Ophthalmic Center, Sun Yat-sen University (Guang Zhou, China). The rats were sacrificed by an intraperitoneal injection of 4% chloral hydrate (Sigma, St. Louis, MO, USA) before the eyes were resected. All efforts were made to minimize suffering.

### 2.2. In Vitro Blue Light Exposure

Primary rat retinal cells were isolated and cultured in accordance with previous described methods [20]. Briefly, approximately four eyes were harvested from one-day old SD rats for each experiment. The retina was isolated and then incubated for 20 min in media containing 0.125% trypsin solution to dissociate the cells. The cells were seeded at a density of 1 × 106 cells/mL in Dulbecco’s modified eagle medium (DMEM, Invitrogen, Carlsbad, CA, USA) with 10% fetal bovine serum (FBS) on a plate pre-coated with poly-l-Lysine, and incubated at 37 °C in an atmosphere of 5% CO_2_ and 95% air. Twenty-four hours later, the culture media was changed to neurobasal media to avoid the unknown influence of FBS to the present study. After a 24-h proliferation period, the retinal cells were maintained in the dark or exposed to white light (900 lux, 1500 lux), or blue light (900 lux, 1500 lux) for 2 h. Afterwards, these cells were transferred to a completely dark incubator for continued incubation. The cells were taken for analysis at indicated time points (2 h, 24 h and 48 h).

The in vitro short-term blue light exposures were achieved by using a light-emitting diode (LED)-based system (Zhaoxin, Nanjing, China) placed in a cellular incubator where the cells were maintained under culture conditions. This system produces a low radiant heat output, avoiding hyperthermic disturbance.

### 2.3. Cell Treatment

To inhibit DNA repair, the retinal cells were treated with DNA-PK inhibitor NU7441 (Tocris Bioscience, Bristol, UK). One hour before light treatment, the retinal neurocytes were pre-treated with NU7441 (1 μM) or the appropriate vehicle control (DMSO).

To inhibit apoptosis, the retinal cells were treated with Z-VAD-FMK (Selleckchem, Houston, TX, USA). One hour before light treatment, the retinal neurocytes were pre-treated with z-VAD-FMK (100 µM) or the appropriate vehicle control (DMSO).

### 2.4. In Vivo Blue Light Exposure

Twenty-four hours before blue light exposure, one-month old SD rats were kept in complete darkness and administrated with 1% tropicamide (Santen, Osaka, Japan) for pupil dilation. Afterwards, the rats were housed in a 12-h light-dark cycle, randomly divided into control and blue light (1500 lux) groups and placed in light-blocking stainless steel boxes. The box temperature was maintained by an electric fan. The DNA DSBs were assessed 2 h after blue light treatment by γ-H2AX immunofluorescence assay Terminal deoxynucleotidyl transferase dUTP nick end labeling (TUNEL) assay evaluated the cell apoptosis of the rat retina upon 2 h blue light exposure. Spectral domain optical coherence tomography (SD-OCT; Heidelberg, Germany) was performed according to the operation manuals one month after blue light exposure.

For the sake of consistency, the number of apoptotic cells and γ-H2AX positive cells was determined from observing the same number of cells from 10 serial sections near the optic nerve.

### 2.5. Immunofluorescence Assay

The retinal cells or tissue slides were fixed with 4% paraformaldehyde for 15 min and immersed for 10 min in 0.1% Triton X-100. The slides were then blocked for 30 min with 10% Normal Goat Serum. Afterward, the slides were incubated overnight at 4 °C with primary antibodies against rabbit γ-H2AX (1:1000, CST, Danvers, MA, USA), mouse anti-microtubule-associated protein-2 (Map2) (1:100, Boster, Wuhan, China), or mouse anti-glia fibrillary acidic protein (GFAP) (1:100, Boster, Wuhan, China). Secondary anti-mouse antibodies (1:500, CST, Danvers, MA, USA) and anti-rabbit antibodies (1:500, CST, Danvers, MA, USA) were added at room temperature and the nuclei were stained with DAPI. Images were captured by fluorescence microscopy (Leica, Buffalo Grove, IL, USA). To study γ -H2AX foci, 100 cells from one cohort of each group were taken.

### 2.6. Cell Counting kit-8 (CCK-8) Assay

The viability of retinal cells was assessed using a cell counting kit-8 (CCK-8) kit (Invitrogen, Carlsbad, CA, USA). A quantity of 1 × 10^5^ primary cultured retinal neurocytes were seeded in 96-well plates. After 24-h adhesion, the retinal cells were treated with white light or blue light illumination for 2 h. The cells were then transferred to a dark environment to proliferate. Twenty-four hours later, the cells were incubated with a CCK-8 agent for 2 h. The absorbance was measured at 450 nm using a fluorescence plate reader (Power Wave XS; BIO-TEK, Winooski, VT, USA). Cell viability was determined based on the optical density ratio of a treated culture relative to an untreated control.

### 2.7. Terminal Deoxynucleotidyl Transferase dUTP Nick End Labeling (TUNEL) Assay

Twenty-four hours after short-term blue light exposure, the cells or retina tissue sections were fixed with 4% paraformaldehyde and then incubated with TUNEL agent for 2-h in the dark. The nuclei were stained with DAPI and images were captured by fluorescence microscopy.

### 2.8. Western Blotting

The retinal cells were treated as previously described and allowed to incubate in a dark environment for 24 h. Then, the retinal cells were washed with phosphate-buffered saline (PBS) and lysed with radioimmunoprecipitation assay (RIPA) buffer supplemented containing a protease inhibitor cocktail. Total protein was extracted by centrifuging the tubes at 4 °C for 15 min at maximum speed to remove debris. Protein samples were loaded onto a sodium dodecyl sulfate/polyacrylamide electrophoresis gel for separation and then transferred onto a nitrocellulose (PVDF) membrane. The membrane was blocked with 5% milk for 1 h and incubated with primary antibody overnight at 4 ℃. Afterwards, the membrane was incubated with a horseradish peroxidase-conjugated goat anti-rabbit secondary antibody (1:10,000, CST, Danvers, MA, USA). The bands were visualized by using an enhanced chemiluminescence detection system (Millipore, Burlington, MA, USA). GAPDH was used as a loading control. The primary antibodies used were as follows: rabbit anti-Ku80 (1:500, Boster, Wuhan, China) and rabbit anti-γ-H2AX (1:1000, CST, Danvers, MA, USA), Mre11(1:200, Boster, Wuhan, China), Ligase IV (1:1000, Proteintech, Wuhan, China), GAPDH (1:10,000, Proteintech, Wuhan, China).

### 2.9. Statistical Analysis

All in vitro experiments were performed at least in triplicate. The data are presented as the mean ± standard deviation (SD). The differences between the means were evaluated using a two-tailed Student’s *t*-test (for two groups), or analysis of variance (ANOVA, for more than two groups). All the calculations and statistical tests were performed using SPSS (version 17.0; SPSS, Chicago, IL, USA). Differences with * *p* < 0.05 were considered statistically significant in all the analyses.

## 3. Results

### 3.1. Exposure to Blue Light Induces Cell Apoptosis in Retinal Neurocytes

Several lines of evidence suggest that blue light may severely impair retinal neurocytes [10,11]. To understand the underlying mechanism, primary retinal neurocytes were cultured in neurobasal medium and then exposed to blue or white light, in a cellular incubator for 2 h. After blue light treatment, the test group cells were transferred to a dark environment (another incubator) where the control cells were cultured separately. Of the retinal neurocytes cultured in neurobasal medium, 91% were positive for Map2, demonstrating the presence of the retinal neuron (Figure 1A). A TUNEL assay was performed to investigate the cytotoxicity induced by both blue and white light exposure (900 lux) in retinal neurocytes (Figure 1B). The rate of apoptosis cells is presented in histograms (Figure 1C). As shown in Figure 1B, few TUNEL-positive cells were observed in the cells cultured in dark or the cells treated with white light.

The same intensity of blue light significantly induces cell apoptosis in the retinal neurocytest (dark: 8.13 ± 1.19, white light: 11 ± 2.53, and blue light: 33.5 ± 5.1, ** *p* < 0.01; Figure 1C). Similarly, the cell viability assay also shows that short-term, white light does not affect the viability of retinal neurocytes (dark: 100%, 900 lux: 98.71 ± 1.9, and 1500 lux: 95.15 ± 3.6, *p* > 0.05; Figure 1D); however, the same amount of blue light exposure (900 lux, 1500 lux) significantly reduces cell viability in an illuminance-dependent manner (dark: 100%, 900 lux: 63.7 ± 11.1%, and 1500 lux: 40.79 ± 4.7%, ** *p* < 0.01; Figure 1E).

### 3.2. Blue Light Induces DNA Double-Strand Breaks (DSBs) in Retinal Neurocytes

Retinal neurons are post-mitotic cells, and thus display genomic instability in the presence of pathological factors [20]. When DNA breaks accumulate, the cells are expected to undergo apoptosis. Indeed, a DNA electrophoresis assay (Figure 2A) shows severe DNA damage at 2 h 900 lux blue light compared to white-light-exposed cells. Moreover, the DNA DSBs were assessed 2 h after blue light treatment by γ-H2AX immunofluorescence assay in retinal neurocytes. As shown in Figure 2B, the expression level of is γ-H2AX notably up-regulated upon 2 h of blue light exposure (900 lux), compared with either dark treatment or white light exposure (900 lux). The relative intensities of the bands are quantified by densitometry and normalized to GAPDH levels, and the average ratio of γ-H2AX to GAPDH in the dark is defined as 1.0. Figure 2C shows that blue light can significantly induce DNA DSBs in retinal neurocytes compared to the cells cultured in dark and white light (for γ-H2AX, dark: 1, white light: 1.08 ± 0.2, blue light: 4.3 ± 0.62, * *p* < 0.05). Consistently, double staining for Map2 and γ-H2AX demonstrates that 2 h 1500 lux white light exposure does not induce DNA DSBs in retinal neurons, while short-term blue light exposure (900 lux) causes DNA DSBs in retinal neurons, which may account for the cell apoptosis (Figure 2D,E). Prominent γ-H2AX foci are observed in nuclei of Map2 positive cells (Figure 2E). These results further confirm that short-term blue light exposure causes remarkable DNA injury.

### 3.3. Retinal Neuron Is More Sensitive to Blue Light Exposure than Glia Cells

The increase of γ-H2AX foci suggests the accumulation of DNA DSBs in primary cultured retinal neurocytes exposed to blue light. To determine the cell type(s) vulnerable to blue light exposure, retinal neurocytes was treated with blue or white light exposure. Afterwards, double-staining assay of γ-H2AX, and cell-specific markers (Map2 for neuron, and GFAP for glia cells, respectively) was performed. As shown in Figure 3, very few retinal neurons (Map2-positives, Figure 3A1,A2) and glia cells (GFAP-positives, Figure 3B1,B2) displayed γ-H2AX foci in the nuclei of cells cultured in dark and white light. However, blue light (900, 1500 lux) notably induced γ-H2AX foci (red) in the nuclei of retinal neurocytes (Figure 3B). After counting the positive cells, our data show that white light does not affect the DNA DSBs of Map2-positive neurons, compared to that of cells cultured in dark (dark: 1.56 ± 0.6%, white light: 2.2 ± 0.6%, *p* > 0.1, Figure 3C). However, a 2-h exposure of 900 lux blue light induces significant γ-H2AX foci formation in retinal neurons, compared to that of white light (white light: 2.2 ± 0.6%; blue 900 lux: 36.4 ± 5.6%, ** *p* < 0.01, Figure 3C). Moreover, the DNA DSBs increase in an illuminance-dependent manner (Figure 3A3,A4). There is significant difference of DNA DSBs in retinal neurons between 900 lux and 1500 lux treatments (900 lux: 36.4 ± 5.6%; 1500 lux: 61.3 ± 6.4%, ** *p* < 0.01, Figure 3C). Contrastingly, glia cells do not show accumulation of DNA breaks after either white light (0 cell) or 900 lux blue light exposure (3.7 ± 2.4%) but show DNA damage at 1500 lux blue light (48.1 ± 14.5%, ** *p* < 0.01, Figure 3D). Moreover, the same phenomena were observed in the retinal cells at 0 min, 10 min, 30 min and 1 h after blue light treatment (Appendix A). The γ-H2AX foci rapidly formed in retinal neurons but not in glia cells upon blue light treatment, demonstrating DNA DSBs. These observations indicate that retinal neurons are more vulnerable to blue light injury, when compared with glia cells.

### 3.4. Ku80 Is Up-Regulated in Glia Cells but not in Retinal Neurons

As we described above, blue light exposure induces DNA DSBs in retinal neurocytes. Which DNA repair signaling is involved in retinal cells upon blue light induced DNA damage? To address this question, we assayed the expression level of mRNA and protein of DNA repair-related genes, such as ligase IV, Mre11, Ku80, by q-RT-PCR and Western blot. As shown in Figure 4A,B, the mRNA expression of Mre11 and ligase IV genes are not significantly different (for Ligase IV, dark: 1, white light: 1.14 ± 0.29, blue light: 1.18 ± 0.1, *p* > 0.05; for Mre11, dark: 1, white light: 1.17 ± 0.1; blue light: 1.05 ± 0.28, *p* > 0.05). Very interestingly, the Ku80 expression is significantly up-regulated in retinal cells treated with blue light, as compared with the cells cultured in the dark and white light (control: 1; white light: 1.06 ± 0.05; blue light: 1.38 ± 0.26, * *p* < 0.05, Figure 4C). The Western blot assays show that protein changes are consistent with mRNA in parallel: the expression of ligase IV and Mre11 remains the same (for Ligase IV, dark: 1; white light: 0.95 ± 0.16; blue light: 1.01 ± 0.14, *p* > 0.05; for Mre11, dark: 1; white light: 0.98 ± 0.26; blue light: 1.03 ± 0.26, *p* > 0.05. Figure 4D,E). Accordingly, Ku80 is up-regulated in retinal cells after exposure of blue light, compared to the cells cultured in the dark (dark: 1; white light: 1.07 ± 0.1; blue: 1.93 ± 0.4, * *p* < 0.05, ** *p* < 0.01. Figure 4D or Figure 4F).

To further confirm the role of DNA breaks in blue light damage, we treated cells with pan-caspase inhibitor, Z-VAD FMK. Our data shows that the caspase inhibitor does not affect the γ-H2AX formation in retinal neurocytes induced by blue light (DMSO: 100%, Z-VAD: 98.3 ± 4.5%, *p* > 0.05, Figure 4G). In contrast, the DNA-PK inhibitor, NU7441, further promotes the γ-H2AX foci formation in retinal neurocytes upon blue light treatment, compared to that of retinal neurocytes treated with vehicle control (DMSO) (DMSO: 100%, NU7411:172 ± 17%, * *p* < 0.05, Figure 4H). This indicates that DNA-PK plays an important role in DNA double strand breaks. Therefore, our data further suggests that blue light induces severe DNA double strand breaks in retinal neurocytes.

More interestingly, by using double immunofluorescence staining assay, we find that Ku80 (red) is absent in Map-positive cells (green) of the primary cultured retinal cells, indicating that Ku80 is mainly expressed in the glia cell (white arrowhead), not retinal neuron (white arrow, Figure 5A) upon blue light treatment. To further confirm that Ku80 is up-regulated in proliferated glia cells, we cultured primary glia cells and retinal neurons, respectively, as shown in Figure 5B. After blue light treatment, all of the whole proteins were extracted and analyzed by Western blot. Our data show that the expression level of Ku80 in glia cells is significantly higher than that in retinal neurons (neurons: 1; glia cells: 2.15 ± 0.35, * *p* < 0.01, Figure 5C,D). This might account for the phenomenon that glia cells are more resistant to blue light-induced DNA damage than retinal neurons. Moreover, the blue light exposure significantly impairs the cell viability of retinal neurons in an illuminance-dependent manner (dark: 100%, 900 lux blue light: 80 ± 9.5%, 1500 lux 61.7 ± 7.6%, * *p* < 0.01, Figure 5E), but did not affect the cell viability of glia cells (Dark:100%, 900 lux blue light:94 ± 2.6%, 1500 lux 91.7 ± 4.6%, *p* > 0.05, Figure 5F).

### 3.5. Blue Light Induces Damages in Retina In Vivo

To explore blue light-induced injury further, an in vivo rat model was established. After 2 h of 1500 lux blue light exposure, the rat eyes were removed immediately. The double staining assay demonstrated that more γ-H2AX foci (red) formed in the ganglion cell layer (GCL) of the rat retina upon blue light treatment, compared with that of rats housed in a normal 12-h light-dark cycle, indicating a significant accumulation of DNA DSBs (control: 2 ± 0.57, blue light: 35.3 ± 4.33, ** *p* = 0.002, Figure 6A,B). More importantly, distinct γ-H2AX foci are detected in most of the Map2-positive cells (green, arrow), and not in GFAP-positive cells (green, arrowhead), in the GCL of rat retina following 2 h of blue light exposure (Figure 6C). These results further suggest that retinal neurons are more sensitive to blue light-induced damage than glia cells due to DNA DSBs. Moreover, the TUNEL-positive cells (white arrows) are observed in the GCL and inner nuclear layer (INL) of the retina after 2 h of blue light treatment (Figure 6D). We analyzed blue light-induced changes in the integrity of the rat retina by measuring retinal thickness with Spectral domain optical coherence tomography (SD-OCT). As shown in Figure 6E,F, a month-long cycle of blue light exposure significantly thinned the retina, compared to controls (Control: 266.5 ± 6.38, blue light: 227.1 ± 9.7, * *p* < 0.001). These observations demonstrate that blue light exposure can induce severe retinal damage in vivo.

## 4. Discussion

Irreversible DNA damage generally causes cell apoptosis [22]. This study demonstrates that even a short-term exposure to blue light can induce a pronounced accumulation of DNA breaks, thereby suppressing cell viability in retinal cells, both in vitro and in vivo. Moreover, we found that non-proliferative retinal cells, retinal neurons, were more vulnerable to DNA breaks induced by blue light exposure than proliferative cells, glia cells, which might be related to DNA DSBs. This was also confirmed in vivo. After analyzing the expression level of proteins related to DNA repair, we find that Ku80 is highly expressed in glia cells compared to retinal neurons. Thus, the current study provides new insights into the mechanisms of blue light induced damage in retinal cells.

Blue light is characterized as high-energy irradiation that can reach the retina and cause irreversible cellular injury [23,24,25,26]. Here, we demonstrated that 2 h of blue light exposure can induce severe cellular injury in a light-intensity dependent manner: suppressing cell viability and inducing apoptosis, compared to dark or white light (Figure 1). This is consistent with previous studies, which demonstrated the apoptotic effect induced by long-term exposure of this short wavelength light in RGC-5 and R28 cell lines [9,15,16]. Moreover, animal studies suggested that retinal degeneration is accelerated by increased light levels [21,27,28,29]. Together, this evidence suggests that wavelength plays a more decisive role than light intensity in retina.

As a part of the central neuron system, retinal neurons are known for their genetic instability and deficiencies in DNA repair [29]. Here, our data show an accumulation of DNA breaks in retinal neurons 2 h after blue light exposure at 900 lux (Figure 2). This is consistent with other studies. For example, Gordon et al. demonstrated light-induced DNA damage in photoreceptors in vitro [22]. Furthermore, lutein attenuated light-induced DNA damage in mice and was able to alleviate the degeneration of photoreceptors and rescue visual function [21]. DNA damage includes different types, such as, single strand damage (SSD) and DSBs. Our data show that γ-H2AX, a marker of DNA double-strand breaks (DSBs), significantly increased in retinal neurons treated with blue light, compared with white light (Figure 2B). It should be noted that H2AX phosphorylation is induced by not only by DNA damage but also apoptotic signaling [30,31]. Accordingly, our data show that the apoptosis inhibitor (Z-VAD) does not affect the γ-H2AX foci formation in retinal neurocytes upon blue light exposure, indicating that blue light treatment induces significant DNA DSBs in retinal neurons.

In addition, we find post-mitotic retinal neurons are more sensitive to blue light exposure than glia cells. Blue light at 900 lux induced γ-H2AX foci formation in retinal neurons, but the effect was not as dramatic in glia cells, suggesting a selective vulnerability of retinal neurons (Figure 3). Histone H2AX phosphorylation is an early signaling event triggered by DNA double-strand breaks (DSBs) [32]. The difference of the γ-H2AX foci upon blue light exposure can be due to many reasons including difference in repair efficiency, difference in repair pathway choice, difference in γ-H2AX expression baseline and difference in sensitivity to blue light [33]. In this study, we investigated the γ-H2AX expression in retinal neurocytes in the retinal cells 0 min, 10 min, 30min, 1 h (Appendix A) and 2 h (Figure 3) after light treatment. Distinct γ-H2AX foci rapidly formatted in retinal neurons (Appendix A, 0 min), but not in glia cells (Appendix A). The same phenomena were observed in the retinal cells at 10 min, 30 min, 1 h (Appendix A) and 2 h (Figure 3) after blue light treatment, demonstrating that the difference in sensitivity to blue light might account for the difference of the γ-H2AX foci formation in retinal neurons and glia cells. However, the different DNA repair capacity might also contribute to the phenomenon that glia cells are more resistant to blue light-induced DNA damage than retinal neurons.

It is known that neurons are more prone to various injures such as ischemia and radiation treatment due to their weak DNA repair ability, compared with glia cells [34,35]. We analyzed the expression level of genes related to DNA DSBs repair. Ligase IV and Mre11 are not changed in retinal neurocytes after treatment with blue light. However, Ku80 is highly expressed in proliferated glia cells (Figure 4I or Figure 5). Ku80 plays an important role in DNA DSBs in cells by binding to DSB ends and is required for the non-homologous end joining (NHEJ) pathway of DNA repair [36,37]. Our data show that the DNA-PK inhibitor (NU7441), which would suppress NHEJ in cells, significantly increased γ-H2AX expression in retinal neurocytes cultured in blue light, indicating that NHEJ might play an important role in the DNA repair signaling of blue light damage. Therefore, these results theoretically elucidate that retinal neurons are more sensitive to blue light exposure than glia cells due to the failing of DSBs repair. However, its underlying mechanism needs further investigation.

In conclusion, this study shows that short-term exposure to blue light immediately causes DNA double-strand breaks and subsequent cellular damage to retinal neurons, in an illuminance-dependent manner, both in vivo and in vitro. Moreover, retinal neurons are more vulnerable to blue light damage than glia cells, which might be caused by different DNA repair mechanisms. Collectively, this study provides not only a new insight into the early pathogenesis of blue light-induced cellular damage in retinal cells, but also the theoretical basis for preventing photochemical damage induced by blue light.

## Figures and Tables

**Figure 1 cells-08-00068-f001:**
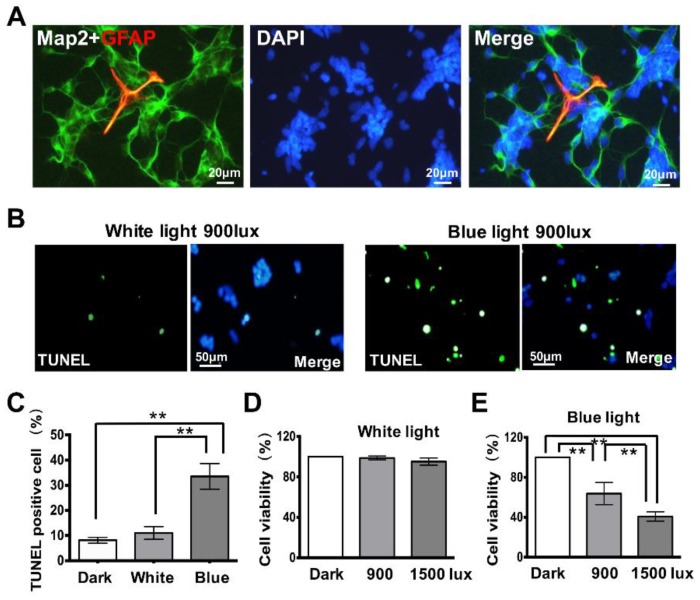
Blue light reduces the viability of retinal neurocytes. (**A**) Double staining for Map2 and glia fibrillary acidic protein (GFAP) in primary cultured retinal neurocytes. (**B**) Terminal deoxynucleotidyl transferase dUTP nick end labeling (TUNEL) assays show blue light exposure induces apoptosis in retinal neurocytes as represented by increased green markers. (**C**) The apoptosis cell number is presented as histogram. (**D**) White light exposure for 2 h at 900 lux or 1500 lux did not affect viability of retinal neurocytes. (**E**) Blue light exposure for 2 h at 900 lux or 1500 lux reduced viability of retinal neurocytes in an illumination-dependent manner. Error bars represent mean ± SD. Asterisks indicate statistically significant differences between control and experimental samples (** *p* < 0.01).

**Figure 2 cells-08-00068-f002:**
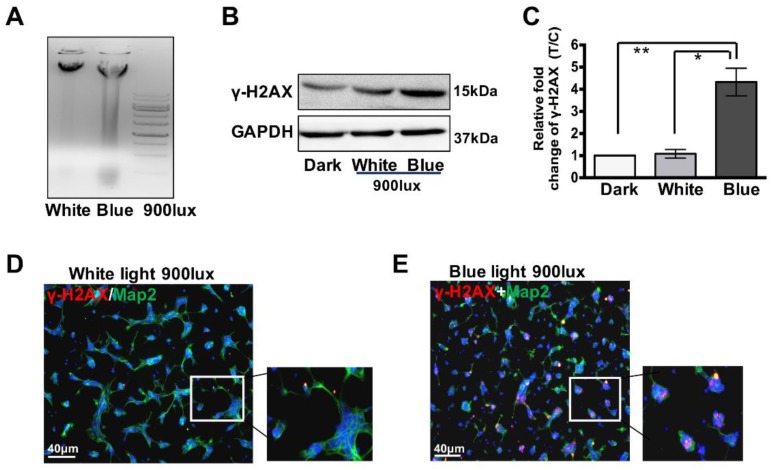
Blue light induces DNA damage in retinal neurocytes. (**A**) A DNA electrophoresis assay shows that severe DNA damage. (**B**) Western blot analysis shows 2 h blue light exposure triggers the up-regulation of γ-H2AX. (**C**) Protein expression was quantified by densitometry, showing the significant up-regulation of target proteins following blue light exposure. (**D**) White light at 1500 lux white light did not induce γ-H2AX foci formation in primary cultured retinal cells. (**E**) Increased γ-H2AX foci forms in retinal neurocytes after blue light exposure. Error bars represent mean ± SD. Asterisks indicate statistically significant differences between control and experimental samples (* *p* < 0.05, ** *p* < 0.01).

**Figure 3 cells-08-00068-f003:**
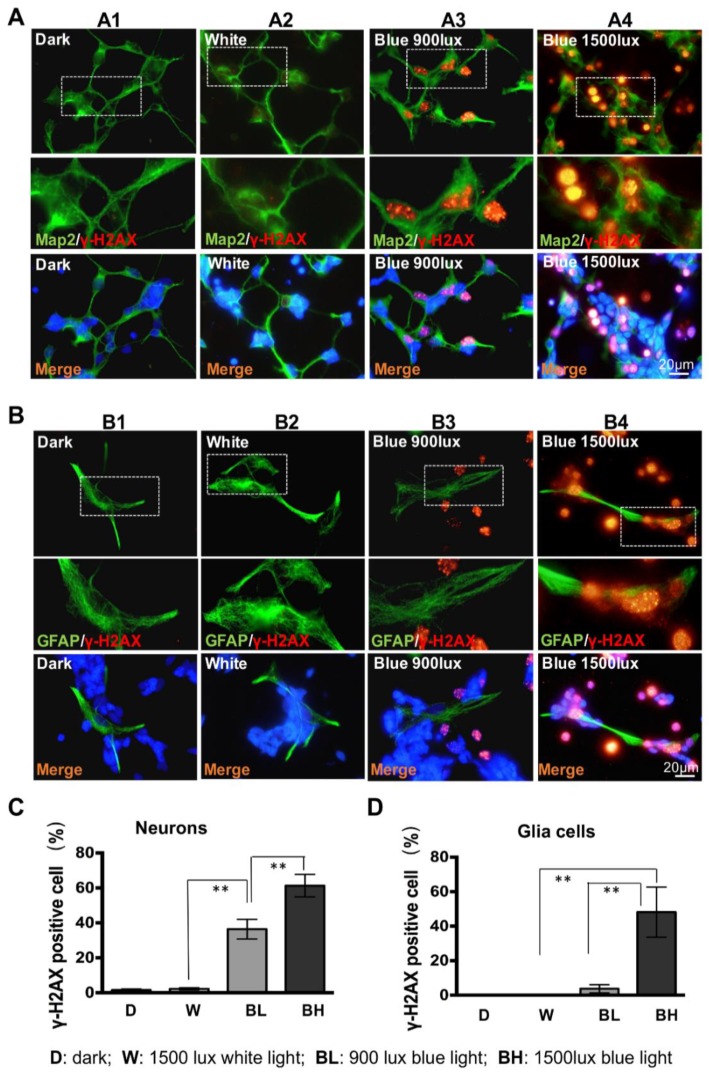
Retinal neurons are more vulnerable than glia cells following blue light exposure. (**A**) Double staining for Map2 and γ-H2AX in neurocytes with or without light exposure. (**B**) The γ-H2AX positive retinal neurons cell numbers are presented as histogram. (**C**) Double staining for GFAP and γ-H2AX in glia cells with or without light exposure. (**D**) The γ-H2AX positive retinal glia cell numbers are presented as a histogram. The magnification of the γ-H2AX foci of the white dotted box area are represented. Error bars represent mean ± SD. Asterisks indicate statistically significant differences between control and experimental samples (** *p* < 0.01).

**Figure 4 cells-08-00068-f004:**
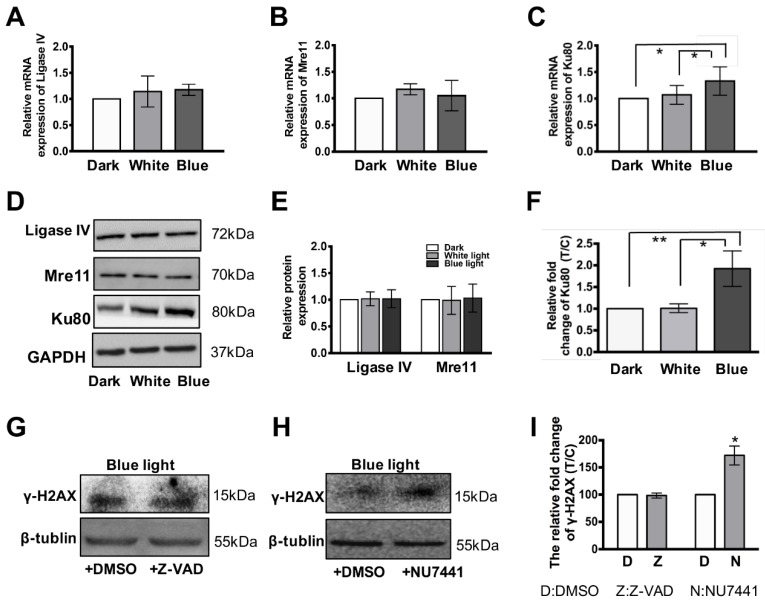
Blue light up-regulated Ku80 expression in glia cells. (**A**) Both blue light and white light treatment do not affect the mRNA expression of ligase IV in primary cultured retinal cells. (**B**) Both blue light and white light treatment do not affect the mRNA expression of Mre11 in primary cultured retinal cells. (**C**) The mRNA expression of Ku80 is significantly up-regulated in primary cultured retinal cells upon blue light exposure, compared with control and white light treatment. (**D**) The Ku80 protein expression is significantly up-regulated in primary cultured retinal neurocytes upon blue light exposure, compared with control and white light treatment. While the light treatment does not affect the protein expression of ligase IV and Mre11. (**E**) The relative protein expression of ligase IV and Mre11 are presented as histogram. (**F**) The relative protein expression of Ku80 were presented as a histogram. (**G**)The pan-caspase inhibitor, Z-VAD FMK did not affect the γ-H2AX formation in retinal neurocytes c induced by blue light. (**H**) The DNA repair inhibitor, NU7441, further promoted the γ-H2AX foci formation in retinal neurocytes upon blue light treatment. (**I**) The relative protein expression γ-H2AX is presented as a histogram. Error bars represent mean ± SD. Asterisks indicate statistically significant differences between control and experimental samples (* *p* < 0.05, ** *p* < 0.01).

**Figure 5 cells-08-00068-f005:**
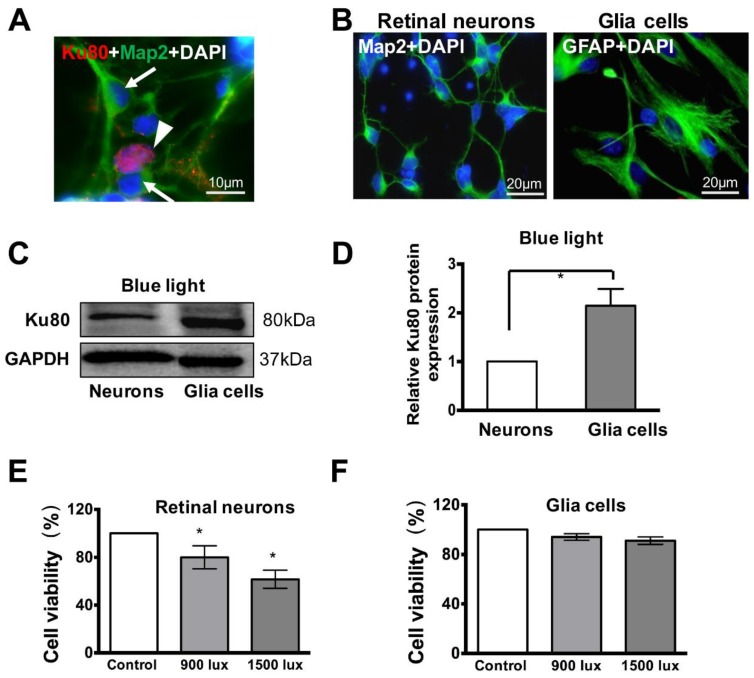
Ku80 is more up-regulated in glia cells than that in neurons. (**A**) Ku80 is more up-regulated in retinal glia cells other than retinal neurons. (**B**) The retinal neurons and glia cells are labelled with Map2 and GFAP, respectively. (**C**) Western blot assay demonstrates that Ku80 is more up-regulated in glia cells than that in neurons upon blue light treatment. (**D**) The relative protein expression of Ku80 in glia cells and neurons were presented as a histogram. (**E**) The blue light impairs the cell viability of retinal neurons. (**F**) The blue light does not affect the cell viability of glia cells. Error bars represent mean ± SD. Asterisks indicate statistically significant differences between control and experimental samples (* *p* < 0.05).

**Figure 6 cells-08-00068-f006:**
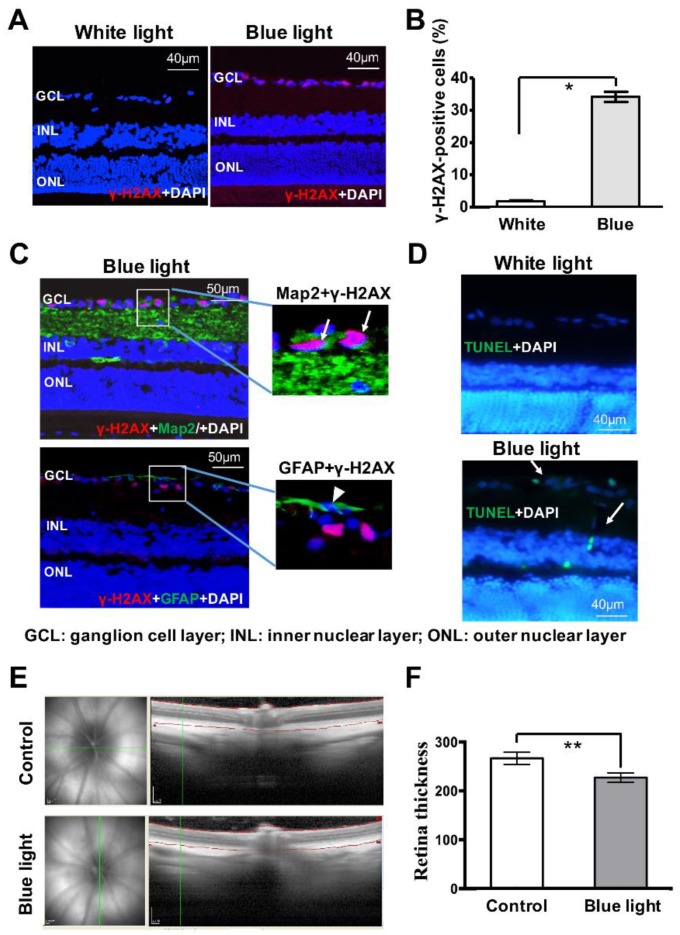
Short term blue light induces DNA damage in retinal neurocytes in vivo. (**A**) γ-H2AX (red) staining demonstrates that blue light induces significant DNA double-strand breaks (DSBs) in GCL of rat retina compared with that of white light. (**B**) The γ-H2AX positive retinal neurocytes cell numbers were presented as a histogram. (**C**) Double staining assay of Map2 (green) or GFAP (green) and γ-H2AX (red) indicates that 2 h of blue light exposure induces γ-H2AX foci formation in Map2-positive cells in rat retina. (**D**) Blue light exposure induced apoptosis in rat retina. (**E**) Histogram representing the significant difference of retina thickness of rat with or without blue light exposure for a month. (**F**) One month of blue light exposure induced retina thinness in the rat model, as evidenced by SD-OCT. (GCL: ganglion cell layer; INL: inner nuclear layer; ONL: outer nuclear layer). Error bars represent mean ± SD. Asterisks indicate statistically significant differences between control and experimental samples (* *p* < 0.05, ** *p* < 0.01).

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
