# Peer review of "Retinal Neuron Is More Sensitive to Blue Light-Induced Damage than Glia Cell Due to DNA Double-Strand Breaks"

_cells, 2019, doi:10.3390/cells8010068_

Round 1

Reviewer 1 Report

The manuscript by Chen, et al. provide impressive data against recent blue light issues by using in cellular and animal models.

I think that their manuscript will be infromative for readers of this journal.

Before publication, the authors should be considered following issues.  

In figure 2, the authors showed the expression of gammaH2AX in cells exposed with white or blue light. It should be noted that H2AX phosphorylation is induced by not only by DNA damage but also apoptotic signaling, which is dependent on caspase-3 activation and following DNA-PK signaling. The authors demonstrated that the blue light exposure increased the percentage of TUNEL-positive apoptotic cells, thus indicating the increased gammaH2AX in WB/ICC may reflects the increased apoptotic cells (Casp3 and DNA-PK activation) . In considering these factors, I cannot judge whether the data shown by the authors really reflect the effect of blue light on DNA double strand breaks, but not apoptotic signaling.

If the authors want to diminish the influence of apoptotic signaling, pan-caspase inhibitors, such as Z-VAD FMK, supports their conclusion at least in vitro. In addition, gammaH2AX foci should be measured by high magnification and high resolution of microscopic data if the authors insist on the increased intensity of gammaH2AX is depends on blue light exposure (TRAIL-induced apoptotic signaling has already shown that the pan-phospho-H2AX signaling in the absence of DNA damage). 

In the provided figures, I cannot recognize clear gammaH2AX foci in figures provided by the authors. 

Ku80 expression was increased by Blue light exposure but the significance of elevated expression of Ku80 is missing. I’m wondering whether the Non-homologus endjoining (NHEJ) is really important for the repair of DNA damage induced by Blue light. NHEJ deficient model (such as SCID(Prkdc-/-), Ku70/80 deficient or RAG1 or 2 deficient model) will support your conclusion. If you cannot use this model, DNA-PK inhibitors such as NU7026/7441 or Lig4 inhibitor (SCR-7) will help your experiments.

Author Response

54 The Xianlienan Road,

Guangzhou, P, R China 510060

December 25, 2018

Dear Sir/Madam,

Thank you for giving us an opportunity to revise our manuscript. We appreciate your efforts and constructive comments to improve the quality of our manuscript titled "Retinal Neuron is more Sensitive to Blue Light-induced Damage than Glia cell Due to DNA Double-strand Breaks" (cells-409296). We have carefully considered the comments and reanalyze our data according to reviewers’ advices. The manuscript has been revised accordingly. Our responses to the comments from reviewer are below.

Here within enclosed is our revised manuscript. And the responses to the comments are attached below.

Reviewers' comments:
Reviewer #1: The manuscript by Chen, et al. provide impressive data against recent blue light issues by using in cellular and animal models.

I think that their manuscript will be infromative for readers of this journal.

Before publication, the authors should be considered following issues.  

In figure 2, the authors showed the expression of gammaH2AX in cells exposed with white or blue light. It should be noted that H2AX phosphorylation is induced by not only by DNA damage but also apoptotic signaling, which is dependent on caspase-3 activation and following DNA-PK signaling. The authors demonstrated that the blue light exposure increased the percentage of TUNEL-positive apoptotic cells, thus indicating the increased gammaH2AX in WB/ICC may reflects the increased apoptotic cells (Casp3 and DNA-PK activation). In considering these factors, I cannot judge whether the data shown by the authors really reflect the effect of blue light on DNA double strand breaks, but not apoptotic signaling.

If the authors want to diminish the influence of apoptotic signaling, pan-caspase inhibitors, such as Z-VAD FMK, supports their conclusion at least in vitro. In addition, gammaH2AX foci should be measured by high magnification and high resolution of microscopic data if the authors insist on the increased intensity of gammaH2AX is depends on blue light exposure (TRAIL-induced apoptotic signaling has already shown that the pan-phospho-H2AX signaling in the absence of DNA damage). 

In the provided figures, I cannot recognize clear gammaH2AX foci in figures provided by the authors. 

Ku80 expression was increased by Blue light exposure but the significance of elevated expression of Ku80 is missing. I’m wondering whether the Non-homologus endjoining (NHEJ) is really important for the repair of DNA damage induced by Blue light. NHEJ deficient model (such as SCID(Prkdc-/-), Ku70/80 deficient or RAG1 or 2 deficient model) will support your conclusion. If you cannot use this model, DNA-PK inhibitors such as NU7026/7441 or Lig4 inhibitor (SCR-7) will help your experiments.

Response: We thank for the reviewer’s valuable suggestions. Histone H2AX phosphorylation is an early signaling event triggered by DNA double-strand breaks (DSBs) [1]. γ-H2AX forms during apoptosis [2]. We treated cells with pan-caspase inhibitor, Z-VAD FMK. Our data shown that the caspase inhibitor does not affect the γ-H2AX formation in retinal neurocytes induced by blue light (Fig. 4G), which indicates that DNA fragmentation can not be inhibited by incubating cultures with caspase inhibitor. In contrast, DNA-PK inhibitor, NU7441, also increased γ-H2AX in retinal neurocytes cultured in blue light (Fig. 4G). DNA-PK plays an important role in non-homologous end joining (NHEJ) pathway. Therefore, our data further suggests that blue light affects DNA double strand breaks, not apoptotic signaling. The manuscript has been revised and it has been highlighted (Line280-288, Line 400-404).

Ku80 plays a key role in NHEJ. In this study, we observe a significant up-regulation of Ku80 in glia cells upon blue light treatment. Moreover, the DNA-PK inhibitor (NU7441), which would suppress NHEJ in cells, significantly increased γ-H2AX expression in retinal neurocytes cultured in blue light. These results indicating that NHEJ might plays an important role in the DNA repair signaling of blue light damage, which might account for the phenomenon that retinal neurons are more sensitive to blue light exposure than glia cells. The manuscript has been revised as reviewer suggested, and it has been highlighted (Line 283-286, Line 414-420)

Moreover, the high magnification results have been added to Figure. 3 as reviewer suggested.

1.    Natale F, Rapp A, Yu W, Maiser A, Harz H, et al. Identification of the elementary structural units of the DNA damage response. Nat Commun. 2017;8:15760. 0.

2.    Emmy P. Rogakou, Wilberto Nieves-Neira, Chye Boon, Yves Pommier, William M. Bonner. Initiation of DNA Fragmentation during Apoptosis Induces Phosphorylation of H2AX Histone at Serine 139. J. BIOLOGICAL CHEMISTRY. 2000; (Vol. 275)13: 9390–9395.

We improved the manuscript to the best of our ability by making certain modifications to the manuscript.

We greatly appreciate the time and effort of the Editor and the reviewers, and we hope that the revised manuscript will be acceptable for publication.

Yours sincerely, 

Jing Zhuang, Ph. D.

Professor, State Key Laboratory of Ophthalmology, Zhongshan Ophthalmic Center,

Sun Yat-sen University, Guangzhou, P. R. China.

Tel: 011-86-20-87330296(Office), 86-13360572251(cell)

Reviewer 2 Report

Blue lights induce DNA damage including DNA breaks. In the study, Chen and colleagues test whether proliferative and non-proliferative retinal cells respond differently to blue light exposure in terms of DNA damage. They concluded that the retinal neurons are more prone to blue light induced DNA damage than the proliferative glia cells. However, the evidence presented is not sufficient to support such claim.

Major concerns:

Authors suggested that the neuronal cells are more sensitive to blue lights based on a single line of evidence that the 2 h exposure of 900 lux blue light does not induced as much gamma-H2AX expression in glia cells in comparison to neuronal cells. However, there is major problem in such interpretation. 

Gamma-H2AX is an indirect marker of DNA breaks. The difference of the gamma-H2AX foci 24 h after exposure can be due to many reasons including difference in repair efficiency, difference in repair pathway choice, difference in gamma-H2AX expression baseline and, of course, difference in sensitivity to blue light. The data here does not distinguish among all the possibilities. In fact, the data hinted at least two of the alternative explanations: the difference in gamma-H2AX expression baseline or repair capability, as shown as the lower % of gamma-H2AX positive cells under dark and white light condition; the difference in repair pathway choice, as suggested by the difference in Ku80 expression. In order address all the possibilities, authors need to include alternative assays that directly examine the formation of DNA breaks such as the TUNEL assay. Also, authors need to include time course experiments to address possible difference in DNA break formation vs. repair efficiency. 

Second, it is based on a single dose. When looking at 1500 lux data or examine the ratio of blue light treatment over dark or white light, one may conclude differently. This again ties into the baseline question and repair pathway choice.  

In addition, since authors are able to culture glia cells and retinal neurons separately, it would be worthwhile to test their survival rates following exposure to blue lights.

Minor concerns:

1.    Line 48. “There is an increasing number of caspase-independent pathways…” this sentence does not make much sense. 

2.    Line 62. “DNA damage … remains unclear”. What about “DNA damage” “remains unclear”. Do you mean mechanism, or sensitivity, or repair? 

3.    Last sentence of the introduction section. This is a descriptive study. Nothing presented or discussed so far provided tangible insight into the “underlying mechanisms”.

4.    Line 164. “control cells were cultured separately”. Were control cells cultured in a separate incubator after exposure? Should not they serve a better control if cultured in the same incubator with the blue light treated samples?

5.    Figure 4I. Please provide more detail to this experiment. 

6.    Language issues. Just to name a few:

Line 237. “there is significantly diffrence”: “significantly” -> “significant”.

Line 252. “Ku80 is up-regulated in glia cells than that in retinal neurons”: “than that”-> “but not”

Line 254. “Whether does….”. Rewrite the sentence in correct grammar. 

Please carefully proofread the manuscript. 

Author Response

54 The Xianlienan Road,

Guangzhou, P, R China 510060

December 25, 2018

Dear Sir/Madam,

Thank you for giving us an opportunity to revise our manuscript. We appreciate your efforts and constructive comments to improve the quality of our manuscript titled "Retinal Neuron is more Sensitive to Blue Light-induced Damage than Glia cell Due to DNA Double-strand Breaks" (cells-409296). We have carefully considered the comments and reanalyze our data according to reviewers’ advices. The manuscript has been revised accordingly. Our responses to the comments from reviewer are below.

Here within enclosed is our revised manuscript. And the responses to the comments are attached below.

Reviewer #2:

Blue lights induce DNA damage including DNA breaks. In the study, Chen and colleagues test whether proliferative and non-proliferative retinal cells respond differently to blue light exposure in terms of DNA damage. They concluded that the retinal neurons are more prone to blue light induced DNA damage than the proliferative glia cells. However, the evidence presented is not sufficient to support such claim.

Major concerns:

Authors suggested that the neuronal cells are more sensitive to blue lights based on a single line of evidence that the 2 h exposure of 900 lux blue light does not induced as much gamma-H2AX expression in glia cells in comparison to neuronal cells. However, there is major problem in such interpretation. 

Gamma-H2AX is an indirect marker of DNA breaks. The difference of the gamma-H2AX foci 24 h after exposure can be due to many reasons including difference in repair efficiency, difference in repair pathway choice, difference in gamma-H2AX expression baseline and, of course, difference in sensitivity to blue light. The data here does not distinguish among all the possibilities. In fact, the data hinted at least two of the alternative explanations: the difference in gamma-H2AX expression baseline or repair capability, as shown as the lower % of gamma-H2AX positive cells under dark and white light condition; the difference in repair pathway choice, as suggested by the difference in Ku80 expression. In order address all the possibilities, authors need to include alternative assays that directly examine the formation of DNA breaks such as the TUNEL assay. Also, authors need to include time course experiments to address possible difference in DNA break formation vs. repair efficiency. 

 Response: We thanks for the reviewer’s valuable suggestions. Histone H2AX phosphorylation is an early signaling event triggered by DNA double-strand breaks (DSBs) [1]. Detection of γH2AX has become the most widely used method for quantification of DSBs and their repair kinetics [2]. Therefore, we examined the γ-H2AX foci in in retinal cells 2 hours after blue light treatment, as described in the method section (Line 116). We are sorry for not describe it more clearly, the manuscript has been revised and it has been highlighted (Line203-204, Line 383).

Moreover, we agree with reviewer that the up-regulation of γ-H2AX in retinal cells might be caused by other reasons. we treated cells with pan-caspase inhibitor, Z-VAD FMK. Our data shown that the caspase inhibitor did not affected the γ-H2AX formation in retinal neurocytes induced by blue light (Fig. 4G). In contrast, the DNA-PK inhibitor, NU7441, further increased the γ-H2AX foci formation in retinal neurocytes upon blue light treatment, compared to that of retinal neurocytes treated with vehicle control (DMSO) (Fig. 4H). DNA-PK plays an important role in DNA double strand breaks. Therefore, our data further suggests that blue light induce severe DNA double strand breaks in retinal neurocytes. The manuscript has been revised and it has been highlighted (Line280-288, Line 400-404).

In addition, we investigate the γ-H2AX expression in retinal neurocyets 1 hours (Supplemental data 1) and 2hours (Fig. 3) after light treatment. Our data demonstrated that distinct γ-H2AX foci formatted in retinal neurocytes 1 hour (Supplemental data 1). and 2 hours (Figure 3) after blue light exposure. While, dark or white light treatment did not induce obvious γ-H2AX foci formation in retinal neurocytes. These data demonstrating that the difference in sensitivity to blue light might account for the difference of the γ-H2AX foci formation in retinal neurons and glia cells. The manuscript has been revised and it has been highlighted (Line250, Line390-398, Line552-558).

Moreover, TUNEL assay has been conducted in our study. As shown in Fig. 1B and Fig. 1C, 24 hours upon treatment, blue light exposed cells show significant cell apoptosis compared to white-light-exposed cells.

1.           Natale F, Rapp A, Yu W, Maiser A, Harz H, et al. Identification of the elementary structural units of the DNA damage response. Nat Commun. 2017;8:15760. 0.

2.           Emmy P. Rogakou, Wilberto Nieves-Neira, Chye Boon, Yves Pommier, William M. Bonner. Initiation of DNA Fragmentation during Apoptosis Induces Phosphorylation of H2AX Histone at Serine 139. J. BIOLOGICAL CHEMISTRY. 2000; (Vol. 275)13: 9390–9395.

Second, it is based on a single dose. When looking at 1500 lux data or examine the ratio of blue light treatment over dark or white light, one may conclude differently. This again ties into the baseline question and repair pathway choice.  

 Response: We thanks for the reviewer’s valuable suggestions. As shown in Fig.3, the γ-H2AX foci has been investigated in retinal neuron and glia cells upon 900lux and 1500lux blue light treatment. Our data shown that, a 2-hour exposure of 900lux blue light induces significant γ-H2AX foci formation in retinal neurons, compared to that of white light. And the γ-H2AX foci formation increases in blue light intensity (Figure 3C, 3D). The 900 lux blue light exposure only induce γ-H2AX foci formation in retinal neurons, while the 1500 lux blue light induce γ-H2AX foci formation in both retinal neurons and glia cells. Therefore, we only examined the expression of DNA repair-related genes in retinal neurocytes upon 900 lux blue light exposure to explore the underlying mechanism of difference light sensitivity between retinal neurons and glia cells.

In addition, since authors are able to culture glia cells and retinal neurons separately, it would be worthwhile to test their survival rates following exposure to blue lights.

 Response: Thanks for the reviewer’s valuable suggestions. The CCK8 assay was performed as reviewer suggested. As shown in Fig.5, the blue light treatment significantly impaired the cell viability of retinal neurons (Dark: 100%, 900lux blue light:80±9.5%, 1500lux 61.7±7.6%, *p<0.01, Fig. 5B), but did not affect the cell viability of glia cells (Dark:100%, 900 lux blue light:94±2.6%, 1500lux 91.7±4.6%, p>0.05, Fig.5C). The manuscript has been revised, and it has been highlighted (Line 311-315).

Minor concerns:

1. Line 48. “There is an increasing number of caspase-independent pathways…” this sentence does not make much sense. 

 Response: We thanks for the reviewer’s valuable suggestions. The inappropriate description has been deleted, and it has been highlighted (Line 48).

2. Line 62. “DNA damage … remains unclear”. What about “DNA damage” “remains unclear”. Do you mean mechanism, or sensitivity, or repair? 

Response: Thanks for the reviewer’s valuable suggestions. The DNA damage sensitivity induced by blue light and its repair mechanism in proliferative cells and terminally differentiated cells remains unclear. The manuscript has been revised as reviewer suggested, and it has been highlighted (Line 60-62).

3. Last sentence of the introduction section. This is a descriptive study. Nothing presented or discussed so far provided tangible insight into the “underlying mechanisms”.

Response: Thanks for the reviewer’s valuable suggestions. The manuscript has been revised as reviewer suggested, and it has been highlighted (Line 64).

4. Line 164. “control cells were cultured separately”. Were control cells cultured in a separate incubator after exposure? Should not they serve a better control if cultured in the same incubator with the blue light treated samples?

 Response: We thanks for the reviewer’s valuable suggestions. We are sorry for the incorrect description. After blue light treatment, the test group cells were transferred to dark environment where the control cells were cultured all alone. The manuscript has been revised as reviewer suggested, and it has been highlighted (Line 171-172).

5. Figure 4I. Please provide more detail to this experiment. 

Response: Thanks for the reviewer’s valuable suggestions. The manuscript has been revised as reviewer suggested, and it has been highlighted (Line 305-308).

6.    Language issues. Just to name a few:

Line 237. “there is significantly diffrence”: “significantly” -> “significant”.

Line 252. “Ku80 is up-regulated in glia cells than that in retinal neurons”: “than that”-> “but not”

Line 254. “Whether does….”. Rewrite the sentence in correct grammar. 

Please carefully proofread the manuscript.

Response: Thanks for the reviewer’s valuable suggestions. The manuscript has been revised as reviewer suggested, and it has been highlighted (Line 244 ,262 and 264). The manuscript has been revised thoroughly by a native English speaker. The English editing certification is attached below.

We improved the manuscript to the best of our ability by making certain modifications to the manuscript.

We greatly appreciate the time and effort of the Editor and the reviewers, and we hope that the revised manuscript will be acceptable for publication.

Yours sincerely, 

Jing Zhuang, Ph. D.

Professor, State Key Laboratory of Ophthalmology, Zhongshan Ophthalmic Center,

Sun Yat-sen University, Guangzhou, P. R. China.

Tel: 011-86-20-87330296(Office), 86-13360572251(cell)

Round 2

Reviewer 1 Report

The manuscript was well improved.

I would like to point out only minor issues as follow.

Line 204, DBSs -> DSBs

LIne 98 & 281,  Z-DEVD-FMK is written in material and methods but Z-VAD FMK is in result and discussion. Which is true?  

Author Response

54 The Xianlienan Road,

Guangzhou, P, R China 510060

January 6, 2019

Dear Sir/Madam,

Thank you for giving us an opportunity to revise our manuscript. We appreciate your efforts and constructive comments to improve the quality of our manuscript titled "Retinal Neuron is more Sensitive to Blue Light-induced Damage than Glia cell Due to DNA Double-strand Breaks" (cells-409296). We have carefully considered the comments and reanalyze our data according to reviewers’ advices. The manuscript has been revised accordingly. Our responses to the comments from reviewer are below.

Here within enclosed is our revised manuscript. And the responses to the comments are attached below.

Reviewers' comments:
Reviewer #1:

The manuscript was well improved.

I would like to point out only minor issues as follow.

Line 204, DBSs -> DSBs

LIne 98 & 281, Z-DEVD-FMK is written in material and methods but Z-VAD FMK is in result and discussion. Which is true?   

Response: We thanks for the reviewer’s valuable suggestions. The manuscript has been revised as reviewer suggested, and it has been highlighted (Line 98 and Line 204).

We improved the manuscript to the best of our ability by making certain modifications to the manuscript.

We greatly appreciate the time and effort of the Editor and the reviewers, and we hope that the revised manuscript will be acceptable for publication.

Yours sincerely, 

Jing Zhuang, Ph. D.

Professor, State Key Laboratory of Ophthalmology, Zhongshan Ophthalmic Center,

Sun Yat-sen University, Guangzhou, P. R. China.

Tel: 011-86-20-87330296(Office), 86-13360572251(cell)

Reviewer 2 Report

I appreciate the authors' effort to address the concerns raised by reviewers. However, I do not believe they have sufficiently answer the question whether the increased gamma-H2AX in retinal neuron is due to higher sensitivity to the blue light or lower capacity to repair DNA damage in general. It would be greatly beneficial to include a parallel gamma-H2AX staining experiment using other DNA damage agents to see if the observation is blue light specific. 

Author Response

54 The Xianlienan Road,

Guangzhou, P, R China 510060

January 6, 2019

Dear Sir/Madam,

Thank you for giving us an opportunity to revise our manuscript. We appreciate your efforts and constructive comments to improve the quality of our manuscript titled "Retinal Neuron is more Sensitive to Blue Light-induced Damage than Glia Cell Due to DNA Double-strand Breaks" (cells-409296). We have carefully considered the comments and reanalyze our data apropos to reviewers’ advices. The manuscript has been revised accordingly. Our responses to the comments from reviewer are below.

Here within enclosed is our revised manuscript. And the responses to the comments are attached below.

Reviewers' comments:
Reviewer #2:

I appreciate the authors' effort to address the concerns raised by reviewers. However, I do not believe they have sufficiently answer the question whether the increased gamma-H2AX in retinal neuron is due to higher sensitivity to the blue light or lower capacity to repair DNA damage in general. It would be greatly beneficial to include a parallel gamma-H2AX staining experiment using other DNA damage agents to see if the observation is blue light specific. 

Response: We thank the reviewer for the valuable suggestions. We assessed the γ-H2AX foci formation in the retinal cells 0 minutes, 10 minutes, 30minutes and 1hour after blue light treatment to address possible differences in DNA break formation vs. repair efficiency. As shown in the Figures below, our data demonstrated that distinct γ-H2AX foci rapidly formatted in retinal neurons (Supplemental data 1, 0 minutes), but not in glia cells (Supplemental data 2). And the same phenomena were observed in the retinal cells 10 minutes, 30minutes, 1hour (Supplemental data 1 and Supplemental data 2) and 2 hours (Figure 3) after blue light treatment. Moreover, the different Ku80 expressions in retinal neurons and glia cells upon blue light treatment demonstrating that the different DNA repair capacity might also contribute to the phenomenon that glia cells are more resistant to blue light-induced DNA damage than retinal neurons. The manuscript has been revised as reviewer suggested, and it has been highlighted (Line 410-412).

As we described in our manuscript, numerous evidence has indicated that neurons are more prone to various injuries such as ischemia and radiation treatment due to its weak DNA repair ability, compared with glia cells [1-2]. Thus, the vulnerability of retinal neurons might not be blue light specific.

Moreover, the English language of this manuscript has been further revised thoroughly.

1.     Tanaka K, Nogawa S, Ito D, et al. Activated phosphorylation of cyclic AMP response element binding protein is associated with preservation of striatal neurocytes after focal cerebral ischemia in the rat. Neuroscience. 2000, 100, 345-354.

2.     Blomgren K, Hagberg H. Free radicals, mitochondria, and hypoxia-ischemia in the developing brain. Free Radic Biol Med. 2006, 40, 388-397.

We improved the manuscript to the best of our ability by making certain modifications to the manuscript.

We greatly appreciate the time and effort of the Editor and the reviewers, and we hope that the revised manuscript will be acceptable for publication.

Yours sincerely, 

Jing Zhuang, Ph. D.

Professor, State Key Laboratory of Ophthalmology, Zhongshan Ophthalmic Center,

Sun Yat-sen University, Guangzhou, P. R. China.

Tel: 011-86-20-87330296(Office), 86-13360572251(cell)
